# The Haves and Have-Nots: The Mitochondrial Permeability Transition Pore across Species

**DOI:** 10.3390/cells12101409

**Published:** 2023-05-17

**Authors:** Elena Frigo, Ludovica Tommasin, Giovanna Lippe, Michela Carraro, Paolo Bernardi

**Affiliations:** 1Department of Biomedical Sciences and CNR Neuroscience Institute, University of Padova, Via Ugo Bassi 58/B, I-35131 Padova, Italy; elena.frigo.7@studenti.unipd.it (E.F.); ludovica.tommasin@studenti.unipd.it (L.T.); michela.carraro@unipd.it (M.C.); 2Department of Medicine, University of Udine, Piazzale Kolbe 4, I-33100 Udine, Italy; giovanna.lippe@uniud.it

**Keywords:** mitochondria, permeability transition, ATP synthase, adenine nucleotide translocase, channels, calcium signaling, cell death

## Abstract

The demonstration that F_1_F_O_ (F)-ATP synthase and adenine nucleotide translocase (ANT) can form Ca^2+^-activated, high-conductance channels in the inner membrane of mitochondria from a variety of eukaryotes led to renewed interest in the permeability transition (PT), a permeability increase mediated by the PT pore (PTP). The PT is a Ca^2+^-dependent permeability increase in the inner mitochondrial membrane whose function and underlying molecular mechanisms have challenged scientists for the last 70 years. Although most of our knowledge about the PTP comes from studies in mammals, recent data obtained in other species highlighted substantial differences that could be perhaps attributed to specific features of F-ATP synthase and/or ANT. Strikingly, the anoxia and salt-tolerant brine shrimp *Artemia franciscana* does not undergo a PT in spite of its ability to take up and store Ca^2+^ in mitochondria, and the anoxia-resistant *Drosophila melanogaster* displays a low-conductance, selective Ca^2+^-induced Ca^2+^ release channel rather than a PTP. In mammals, the PT provides a mechanism for the release of cytochrome *c* and other proapoptotic proteins and mediates various forms of cell death. In this review, we cover the features of the PT (or lack thereof) in mammals, yeast, *Drosophila melanogaster*, *Artemia franciscana* and *Caenorhabditis elegans*, and we discuss the presence of the intrinsic pathway of apoptosis and of other forms of cell death. We hope that this exercise may help elucidate the function(s) of the PT and its possible role in evolution and inspire further tests to define its molecular nature.

## 1. Introduction

A mitochondrial permeability increase leading to swelling [1,2] and its dependence on matrix Ca^2+^ accumulation was described early in mitochondrial research [3]. The term “permeability transition” (PT) to define this event was introduced in 1976 [4] and further defined in 1979 [5,6,7]. With its estimated radius of 14 Å, exclusion size of 1500 Da, lack of selectivity and detrimental consequences on energy conservation [3], the PT was long considered an in vitro artifact or a terminal event in cell damage [8]. This view is understandable because mitochondrial research was developing within the frame of chemiosmotic principles (leading to the Nobel Prize to Peter Mitchell in 1978), which posited that the inner membrane had to be impermeable to solutes and charged species in general [9]. The mechanistic bases for the PT have long been debated in the bioenergetics community. An early hypothesis was that the process involved the generation of long-chain fatty acid(s) and lysophospholipids by Ca^2+^-activated phospholipase A_2_, an idea that is supported by the inducing effects of fatty acids [10] and by the remarkable protection afforded by nupercaine [11] and N-ethylmaleimide [12]. The alternative hypothesis—that the PT could rather be due to the opening of a regulated inner mitochondrial membrane (IMM) channel, the PT pore (PTP) [5,6,7]—was mostly met by skepticism. The channel hypothesis gained ground with the electrophysiological identification of an IMM high-conductance channel (the mitochondrial multiconductance channel [13] or megachannel [14]), which shares key features with the PTP [15,16,17,18] and with the discovery that the PT could be inhibited by nanomolar concentrations of cyclosporin A (CsA) [19,20,21,22] without inhibition of phospholipases [23]. Inhibition by CsA was critical to establish a causal role for the PTP in cell and organ injury [24,25,26,27,28] and in apoptosis [29,30,31]. CsA affects the PTP through inhibition of a matrix peptidyl prolyl *cis-trans* isomerase [22], later shown to be a mitochondrial cyclophilin (CyP), CyPD [32,33]. CyPD favors the PT [32], as shown by genetic ablation of the CyPD-encoding *Ppif* gene, which desensitizes the PTP to Ca^2+^ just like treatment with CsA [34,35,36,37]. CyPD interacts with both the ANT [38] and with the OSCP subunit of F-ATP synthase [39,40]. These interactions are deemed important for the transition of these proteins into Ca^2+^-dependent, high-conductance channels mediating the PT [40,41,42].

## 2. Adenine Nucleotide Translocase and Formation of the Permeability Transition Pore

Adenine nucleotide translocases (ANTs) belong to the SLC25 mitochondrial carrier family, proteins that transport metabolites, inorganic ions and cofactors [43] through a common alternating-access mechanism [44]. ANTs are 30 kDa proteins that catalyze the exchange of ADP and ATP across the IMM [45,46]. Humans possess four isoforms (ANT1-4), with ANT4 being specific to germline and pluripotent stem cells [47,48]. The ANT proteins contain three homologous domains of about 100 amino acids, each comprising an odd-numbered transmembrane α-helix (H1, H3, H5), a loop with a short matrix α-helix (h12, h34, h56), which lies in the plane of the IMM, and an even-numbered set of transmembrane membrane α-helices (H2, H4, H6). This basic structural fold was confirmed for the yeast isoforms Aac2p and Aac3p [49] and for ANT1 in *Bos taurus* [50]. During the catalytic cycle the protein switches from the c-conformation (nucleotide-binding site facing the cytosol) to the m-state (nucleotide-binding site facing the matrix) triggered by ADP binding (Figure 1). The core elements composed of the cytoplasmic ends of the H1, H3 and H5 helices move outward as rigid bodies, opening up the substrate-binding site to the matrix, while the cytoplasmic gates formed by the H2, H4 and H6 helices rotate inwards, closing the cytoplasmic side. The reverse movements occur in the m- to c-state transition triggered by ATP [51,52]. Participation of the ANT in the PT was proposed early, based on the opposing effects of the selective inhibitors atractylate (which favors PTP opening) and bongkrekate (which favors PTP closure) [5]. Channel activity matching several features of the PTP has been reported for ANT from both bovine heart and *Neurospora crassa* reconstituted in giant liposomes [41,42]. Questions were raised by a study of the Wallace laboratory, in which ANT1 and ANT2 were genetically ablated in the mouse liver [53]. Mitochondrial respiration could not be stimulated by the addition of ADP, consistent with the lack of compensation by other isoforms of ANT, but a PT could still occur [53]. PTP opening required higher loads of matrix Ca^2+^ and was potentiated by diamide and *tert*-butyl hydroperoxide but not by atractylate, while it maintained sensitivity to CsA, leading to the conclusion that the ANT is not essential for the mitochondrial PTP [53]. A recent study in liver mitochondria from ANT1/ANT2/ANT4 deficient mice (lacking all ANT isoforms) found a striking resistance to Ca^2+^, and yet these mitochondria underwent a CsA-sensitive PT indicating that ANTs are involved in the process but also that additional permeabilization pathways must exist [54].

As suggested by early work in mitochondria [55,56], recent electrophysiological data have shown that a sizeable fraction of the H^+^ leak (which accounts for basal respiration) takes place through the ANT, which acts as a channel in which fatty acids mediate H^+^ transport without being translocated [57]. These studies demonstrate that the ANT can act as a channel, but this is highly selective for H^+^. Although also PTP opening is favored by fatty acids, there is currently no obvious mechanism to explain how the ANT can form a high-conductance channel. The protein does not possess Ca^2+^-binding sites, suggesting that additional factors may be involved such as cardiolipin [58] or that post-translational modifications may be critical [59]. Ca^2+^-dependent proteolysis should also be considered, given that deletion of amino acids 261–269 in uncoupling protein 1 (an SLC25 family member closely related to ANTs) converts this H^+^-selective channel into a pore, allowing permeation of species with molecular mass up to 1 kDa [60]. It has been suggested that the PT is favored by the oxidation of ANT residues C57 [61,62] and C160 [62], resulting in the formation of disulfide bridges and enhanced binding of CyPD [62], a hypothesis that has not been tested yet by site-directed mutagenesis.

## 3. F-ATP Synthase and Formation of the Permeability Transition Pore

Mitochondrial ATP synthase is a multisubunit complex of about 600-kDa organized in two main domains, the spherical, catalytic F_1_ sector protruding into the matrix and the F_O_ sector embedded in the membrane, firmly connected through a central and a peripheral stalk (Figure 2). The F_1_ sector is an assembly of three αβ subunit pairs organized around the central stalk (subunits γ, δ and ε) in close contact with the F_O_ sector composed of the c-ring, a barrel-shaped structure filled with lipids and a variable number of identical c subunits [63], and of subunit a, which lies in close contact with the c-ring. On the matrix side, the peripheral stalk begins at subunit OSCP, the N-terminus of which forms tight interactions with the “crown region” of the α_3_β_3_ complex [64,65], while its C-terminus connects with subunits F6 and b, which contacts subunit d and reaches into the inner membrane. Within the membrane, the two transmembrane helices of subunit b make a remarkable “U-turn” forming a compact triple transmembrane bundle with helices of subunits e and g through GXXXG motifs. This creates a “hook apparatus” allowing the C-terminus of subunit e to reach out and attach to the c-ring lipids from the intermembrane side [64,65]. Subunits e, f, g, A6L, j and k are involved in dimer formation [66]. 

The first hint that the F-ATP synthase could be involved in the PT was the demonstration that, in the presence of Pi, CyPD interacts with the peripheral stalk at subunit OSCP resulting in partial inhibition of the rate of ATP synthesis and hydrolysis; CyPD binding could be inhibited by CsA [39,40]. The OSCP subunit turned out to be an important site of regulation for PTP formation through the binding of CyPD and its small-molecule mimic Bz-423 [40], as well as through a variety of effectors such as 17β-estradiol [67], honokiol [68], Hsp90 [69], p53 [70], β amyloid protein [71], SIRT3 [72,73] and TRAP1 [74] (see [75] for review).

Clear evidence has been obtained that Ca^2+^-dependent channel formation from F-ATP synthase can actually occur [40,76,77,78]. A plausible mechanism for channel generation was initially proposed by Christoph Gerle as the “death finger” hypothesis [79,80]. This mechanism combines elements from two previous hypotheses, i.e., that the PTP forms in dimers of F-ATP synthase as a result of a conformational change originating at OSCP [40], or in the c-ring after dissociation of F_1_ [76,81]. It can be summarized as follows:

Ca^2+^ binding to the β subunit would cause an increase in the rigidity of the F_1_ sector [82]. This causes mechanical stress on OSCP, which would be transmitted to the peripheral stalk and relayed into the IMM [83,84] at the “wedge” or “bundle” region where subunit e is located [64,65]. As a result, subunit e would exert a pulling effect on the outer lipids of the plug of the c-ring with the formation of a channel within the c-ring itself [64]. At physiological levels of matrix Ca^2+^, the PTP oscillates between closed and open states. When matrix Ca^2+^ increases, channel openings become more stable and favor the eventual displacement of the inner lipids and of the central stalk. Irrespective of the mechanism, the fully open state is reversible when matrix Ca^2+^ is removed [85].

This model is consistent with recent cryo-EM structures of F-ATP synthase prepared in the presence of 5 mM Ca^2+^ [64]. How the channel would form within the c-ring [76] remains an open question, and it is not clear whether all conductance substates can be explained by the above hypothesis. In patch-clamp experiments with native membranes and in highly purified dimeric F-ATP synthase preparations reconstituted in lipid bilayers the PTP exhibits substates ranging from 45 to above 1000 pS [13,14,77]. An interesting possibility is that a channel could form at the monomer-monomer interface, perhaps contributing to the reversible, transient flickering of the PTP observed both in isolated mitochondria and intact cells [13,14,86,87,88]. Some support for this hypothesis comes from the high-resolution reconstruction of the dimer interface based on the monomers, which shows a cavity apparently not filled with lipids between adjacent subunits j [65], and the recent discovery of Mco10, a protein with high similarity to subunit k that regulates the permeability transition in yeast [89].

## 4. Cooperation between F-ATP Synthase and ANT in Formation of the Permeability Transition Pore

Ablation of individual subunits of F-ATP synthase in HAP1 cells prevented the assembly of the complete enzyme complex, with the generation of partially assembled, “vestigial” forms; and yet the PT persisted and maintained its sensitivity to CsA, leading initially to the conclusion that F-ATP synthase does not mediate the formation of the PTP [90,91,92]. It was later shown that HAP1 cells ablated of subunit c still have a channel sensitive to both CsA and bongkrekate, indicating the involvement of ANT, while the channel of wild-type cells is sensitive to CsA only [93]. Our recent work in HeLa cells lacking subunits g and e, in ρ^0^ cells derived from human 143B osteosarcoma and in HAP1 cells lacking subunits b and OSCP indicates that F-ATP synthase and ANT cooperate in PTP formation [94], possibly by physically interacting at the “ATP synthasomes” [95,96,97]. Thus, apparent discrepancies about formation of the PTP from F-ATP synthase can be explained by the existence of two channels modulated by CyPD, one channel being formed from F-ATP synthase and the other from ANT [98].

## 5. The Permeability Transition and Its Role in Mammals

The PT of mammals has been studied very extensively. Here we will highlight specific issues that are relevant to the main topic of the present paper, i.e., an update on species-specific features of the PT [99] that may help understand its function and possible role in evolution. We refer the reader to the still very useful review of Gunter and Pfeiffer for coverage of earlier literature [100], to two thorough reviews on the electrophysiological and general aspects of the pore [17,18] and to a few recent reviews for details on the role of the PT in health, aging and disease [98,101,102,103,104,105,106,107,108] and its pharmacological modulation [109].

The single most important factor required for PTP opening is matrix Ca^2+^ [5,6,7]. Since a PT cannot be observed in its absence [110], Ca^2+^ is best defined as a “permissive” factor. Ca^2+^ is unique because all the other Me^2+^ ions (e.g., Mg^2+^, Sr^2+^ and Mn^2+^) inhibit the PT by competing with Ca^2+^ [16]. In mammalian mitochondria, Pi potentiates the PTP-inducing effects of Ca^2+^ in spite of its lowering effect on matrix free [Ca^2+^] [111], probably a consequence of enhanced CyPD binding [39,112] and possibly of decreased free matrix [Mg^2+^].

The primary consequence of PTP opening is depolarization, while further effects depend on the duration of the open state. For brief openings, which occur under physiological conditions [88], PTP closure is followed by mitochondrial resealing and repolarization. Partial swelling may occur for short PTP open times, with widening of the intracristal junctions that otherwise limit the diffusion of the bulk of cytochrome *c* to the intermembrane space [113]. Junction widening contributes to the release of cytochrome *c* through an intact outer membrane via channels formed by activated Bak and Bax [113]. For longer open times, release of matrix Mg^2+^ and nucleotides takes place, which stabilizes the pore in the open conformation and causes respiratory inhibition. Collapse of the proton gradient curtails ATP synthesis and turns mitochondria into consumers of any available glycolytic ATP [114], while diffusion of ions and solutes followed by water leads to matrix swelling and eventually to outer membrane damage with the release of a variety of intermembrane proapoptotic factors such as cytochrome *c*, endonuclease G, AIF and SMAC-DIABLO [115,116,117,118,119]. This latter event is quite important in the context of apoptosis because it provides a mechanism to release intermembrane space proteins that are too large to permeate through the Bax/Bak pore [120] (Figure 3A).

PTP opening has been extensively studied and demonstrated to play a role in a variety of paradigms, including necrotic, apoptotic and necroptotic cell death, and aging [98,102,103,104,105,106,107,108]. Increasing evidence also indicates that transient PTP openings may provide mitochondria with a Ca^2+^ release channel involved in Ca^2+^ homeostasis, as proposed earlier [121,122] and now supported by several studies [123,124,125,126,127,128,129]. Perhaps the strongest evidence that the PTP plays a key role in Ca^2+^ homeostasis in vivo comes from a recent study of the Casari laboratory in hereditary spastic paraplegia caused by altered SPG7, a mitochondrial protein of the AAA-protease superfamily [130]. SPG7 patient fibroblasts and *Spg7* knockout mouse neurons displayed decreased low conductance PTP opening due to CyPD deacetylation by SIRT3, resulting in Ca^2+^ deregulation, a detrimental effect for neurotransmitter vesicle dynamics and synaptic transmission [129]. Treatment of SPG7 patient cells and *Spg7* knockout neurons with Bz-423, a functional mimic of CyPD able to restore normal PTP openings, normalized synaptic transmission and motor performance in *Spg7* knockout mice, providing a potential therapy for this form of spastic paraplegia [129].

## 6. The Permeability Transition and Its Role in Yeast

Whether a bona fide PT occurs in yeast has been both controversial [99,131] and difficult to address because the vast majority of yeast strains do not possess a uniporter able to mediate rapid Ca^2+^ uptake [132]. The problem was overcome by using the ionophore ETH129, which allows electrophoretic Ca^2+^ uptake [133] leading to PT induction, provided that the concentration of Pi is optimized to prevent its inhibitory effects on the PTP [134]. Indeed, at variance from mammalian mitochondria, the PT of yeast is markedly inhibited by Pi [134]. Other notable differences are the lack of inhibition by CsA [133,134,135] and by ablation of the yeast mitochondrial cyclophilin Cpr3 [135]. We have suggested that Cpr3 does not interact with the PTP even in the presence of Pi, which instead in mammalian mitochondria favors the interaction of CyPD with the OSCP subunit of F-ATP synthase to promote PTP opening [40]. It has been reported that the ethanol-induced PT of *Saccharomyces cerevisiae* can become sensitive to CsA and Cpr3 ablation when the substrate is added to agar-embedded cell or mitochondrial suspensions at low concentrations [136], but the mechanistic basis for this effect, and whether it is limited to induction by ethanol, remains to be assessed. Importantly, ANT from *Neurospora crassa* and F-ATP synthase from *Saccharomyces cerevisiae* can form Ca^2+^-activated, high-conductance channels [42,135,137], and gel-purified F-ATP synthase incorporated in lipid bilayers forms channels with the features expected of the PTP [137], leaving little doubt that a PT can take place in yeast as well. A recent mechanistic advance has been the discovery of Mco10, a novel yeast protein similar to subunit k that predominantly associates with monomers and promotes the PT, possibly by fitting in the same position occupied by subunit k and providing tighter packing of F_O_ subunits [89]. Whether a similar protein fulfills the same function in mammals is an exciting possibility that needs to be addressed.

In mammals, mitochondria play a crucial role in cytosolic Ca^2+^ homeostasis through a variety of transport systems [138]. As already mentioned, yeast mitochondria do not possess a mitochondrial Ca^2+^ uniporter (MCU) complex [139] and therefore their potential role in Ca^2+^ homeostasis is usually overlooked. However, the Ca^2+^ electrochemical gradient provides a large driving force for Ca^2+^ accumulation that may explain why yeast mitochondria contain 8–9 ng atoms of Ca^2+^ per milligram protein, which is close to the Ca^2+^ content of rat liver mitochondria [140]. Consistently, electrophoretic Ca^2+^ uptake coupled to H^+^ ejection can be measured in isolated *Saccharomyces cerevisiae* and *Candida utilis* mitochondria when Ca^2+^ is added at concentrations of 1–10 mM [141]. Respiration-driven uptake is observed with Ca^2+^, Sr^2+^ and Mn^2+^ but not with Mg^2+^ [141], suggesting the existence of a low-affinity transport system with a selectivity similar to that of the MCU [138].

Yeast programmed cell death is increasingly recognized as a physiologically relevant event with intriguing analogies with mammalian apoptosis that are particularly striking for the mitochondrial pathway [142]. In yeast apoptosis, which is usually preceded by increased cytosolic [Ca^2+^], the changes occurring in mitochondria are similar to those observed in mammals including cristae remodeling, increased ROS levels, matrix swelling and cytochrome *c* release [142,143,144]. These changes support the hypothesis that the PT plays an important role in yeast programmed cell death [145], as also indicated by a study of yeast spheroblasts [146]. We surmise that when yeast cells are challenged by death stimuli—which cause increased cytosolic [Ca^2+^] and the onset of oxidative stress—mitochondria can accumulate enough Ca^2+^ to undergo the PT in spite of the absence of the MCU. Once PTP opening occurs, cytosolic and matrix Ca^2+^ equilibrate stabilizing the PTP in the open conformation, which is followed by osmotic swelling of the matrix, outer membrane damage and release of intermembrane proteins that participate in the process of cell death by favoring activation of the metacaspase Ycap1 (Figure 3B), as described in detail in a specific review [145].

## 7. The “Permeability Transition” and Its Role in *Drosophila melanogaster*

In a thorough study of mitochondria from *Drosophila melanogaster*, we found that the essential features of Ca^2+^ transport are shared with those of mammalian mitochondria, including the presence of ruthenium red-sensitive electrophoretic Ca^2+^ uptake and of Na^+^-dependent and Na^+^-independent Ca^2+^ release mechanisms. *Drosophila* mitochondria also undergo a process of Ca^2+^-induced Ca^2+^ release with features that set it apart from the PTP [147]. The most notable differences with the mammalian and yeast PTPs are the absence of Ca^2+^-dependent swelling and of cytochrome *c* release even in KCl-based media, which suggests that Ca^2+^-dependent permeabilization is selective for H^+^ and Ca^2+^ [147]. Lack of swelling by uptake of Ca^2+^ cannot be explained by peculiar structural features of *Drosophila* mitochondria because both matrix swelling and cytochrome *c* release could be induced by the addition of the K^+^ ionophore valinomycin or of the pore-forming peptide alamethicin [147]. Interestingly, gel-purified F-ATP synthase of *Drosophila* generates channels with maximal conductance of a mere 53 pS [148], which is consistent with the in situ results described above. Given that the F-ATP synthase is the strongest candidate for PTP formation, the data suggest that the *Drosophila* species has unique, yet undefined structural features that explain these differences. A subsequent study confirmed that Ca^2+^-induced depolarization and Ca^2+^ release in *Drosophila* are not followed by permeabilization to trapped calcein or by swelling unless mitochondria are treated with a Ca^2+^ ionophore or phenylarsine oxide [149], a potent PTP inducer [150]. Under the latter conditions permeabilization was inhibited by CsA or by knocking down either *Cyp-1*, the gene encoding mitochondrial cyclophilin of *Drosophila*, or the ANT1 gene, *SesB* [149].

The properties of the *Drosophila* Ca^2+^-induced Ca^2+^ release system are related to, but different enough from those of the PTP of mammals and yeast to suggest that the *Drosophila* channel performs a different function. Like the mammalian and yeast pore, the *Drosophila* Ca^2+^-induced Ca^2+^ release channel (CrC) requires matrix Ca^2+^ loading and is favored by inner membrane depolarization, thiol oxidation and treatment with millimolar concentrations of N-ethylmaleimide, by Bz-423 and enforced mitochondrial expression of human CyPD [147,148]; like the yeast PTP, it is inhibited by Pi [147]; at variance from both, the *Drosophila* species is insensitive to ADP and, as mentioned above, does not mediate mitochondrial swelling, an arrangement that may fit the features of apoptosis in this organism.

Many of the proteins important for apoptosis in mammals are conserved in *Drosophila*, but the role of mitochondria in *Drosophila* cell death remains controversial [151,152,153,154,155,156,157,158,159,160,161]. *Drosophila melanogaster* possesses two cytochrome *c* proteins (DC3 and DC4), the latter having the highest homology with cytochrome *c* of other species [162]. At variance from mammals, where cytochrome *c* is required for apoptosome formation with Apaf-1 and therefore for caspase 9 activation [163], in *Drosophila* apoptosis, rupture of the outer mitochondrial membrane and release of cytochrome *c* do not occur even if mitochondrial fragmentation is observed [161]. Cytochrome *c* may rather be displayed on the outer membrane [153], although at least partial release has also been observed [156]. Consistent with a non-essential role of cytochrome *c*, silencing of *DC3* and *DC4* did not affect the processing of dApaf, activation of the caspases DRONC and DrICE (Figure 3C) and rate of cell death in vitro and in vivo [154], yet cytochrome *c* may be essential for effector caspase activation and terminal differentiation of sperm [164,165,166].

In the context of the PT, it is relevant that (i) *Drosophila* embryos are very resistant to lack of oxygen, a condition in which they do not die but rather stop growing and developing until oxygen is supplied back [167], and (ii) adult flies can survive anoxic conditions for hours without signs of tissue damage [168]. Ischemia and reperfusion are the most studied pathological conditions that can induce a PT in mammals [101], and it is very tempting to attribute the anoxia resistance of *Drosophila* mitochondria to the lack of a high-conductance PTP.

## 8. The lack of Permeability Transition in *Artemia franciscana* and Other Crustaceans

*Artemia franciscana* belongs to the taxon Crustacea and, like *Drosophila melanogaster,* belongs to the phylum Arthropoda. This salt- and anoxia-tolerant small brine shrimp lives in salt lakes such as the Great Salt Lake, Utah. Under favorable conditions, *Artemia* has an ovoviviparous development from larvae (nauplia) released by females. Under extreme conditions (such as dropping oxygen levels, increased osmolarity and increased population density), females become oviparous and release gastrulae in diapause, a partially arrested developmental state of dormancy with metabolic activity decreased by 97%. When conditions return to be favorable, the development of larvae resumes [169]. *Artemia franciscana* can resist for many years under anoxia by reducing ATP levels and heat production, and this metabolic arrest appears to be due to the reduction of mitochondrial transcription [170]. Hand and coworkers made the startling discovery that mitochondria from *Artemia franciscana* accumulate large amounts of Ca^2+^ without undergoing a PT [171] with the formation of needle- and dot-like electron-dense material rich in calcium and phosphorus [172]. Remarkably, Ca^2+^ could not induce the PT in two other crustaceans, the brown shrimp *Crangon crangon* and the common prawn *Palaemon serratus* [173], in keeping with the original hypothesis that the absence of the PTP could be a general feature of invertebrates (Figure 3D) [171,174]. Similar to *Drosophila*, cytochrome *c* is not required for caspase 9 activation in *Artemia franciscana* extracts [175]. It is tempting to speculate that lack of the PT is part of the mechanisms that impart resistance to anoxia and desiccation, together with “late embryogenesis abundant” proteins and trehalose [176]. It should be mentioned that treatment of *Artemia franciscana* embryos with 20 μM HgCl_2_ in hypotonic solutions opens a CsA-insensitive permeability pathway with a size exclusion limit of 540 Da, the identity of which was not defined [171]. Interestingly, sulfhydryl reagents including mercurials can switch the mitochondrial aspartate/glutamate carrier and the ANT from obligate counterexchange to unidirectional transport [177,178]. Whether the *Artemia* unspecific permeability can be explained by modification of metabolite carriers, and whether it can take place under normal conditions remains to be established.

*Artemia* ANT presents a peculiar feature that differentiates it from its mammalian counterpart, i.e., lack of sensitivity to BKA, a feature that appears to depend on the 198–225 amino acid region [172]. Surprisingly, the expression of *Artemia* ANT in lieu of AAC2 in *Saccharomyces cerevisiae* AAC1-AAC3 KO strain restores BKA sensitivity [179], a finding that still awaits an explanation. Given that both in yeast and mammals the lipid environment is a critical component for the activity of membrane proteins, the influence of *Artemia* ANT on the yeast lipidome was evaluated. It was found that *Artemia* ANT influences the yeast mitochondrial membrane inducing changes in lysolipids [180], which are highly represented in the c-ring.

## 9. The Permeability Transition and Its Role in *Caenorhabditis elegans*

The first caspase was identified in the nematode *Caenorhabditis elegans* [181,182], an organism that has been essential to elucidate the pathways for apoptotic cell death in mammals [183]. In this organism, apoptosis is executed by CED-3 (which corresponds to mammalian caspase 3) after its activation by EGL-1 (a BH3-only protein similar to Bim and Bid), which antagonizes the inhibitory effect of CED-9 (the homolog of Bcl2) by displacing it from CED-4 (which is the equivalent of Apaf-1). CED-4 is then released from the surface of the outer mitochondrial membrane to activate CED-3 [184] (Figure 3E). Cytochrome *c* is not required for activation of CED-3 by oligomers of CED-4, and yet essential proapoptotic factors such as WAH-1 (homolog of AIF) and CPS-6 (homolog of Endo-G) are released from mitochondria to execute cell death [185]. It has been noted that the mechanism through which apoptogenic factors are released from mitochondria is a fundamental question that still awaits an answer [185]. The PT could provide such a mechanism (Figure 3E) as suggested by recent studies related to aging and autophagy in *Caenorhabditis elegans*.

Autophagy is usually required for lifespan extension in model organisms including *Caenorhabditis elegans* [186]. Unexpectedly, lifespan was instead shortened in worms with increased autophagy caused by lack of serum glucocorticoid regulated kinase-1, a major effector of metabolic regulation downstream of TORC2. The effect was due to increased mitochondrial permeability via upregulation of VDAC1 and PTP activation, as indicated by a variety of criteria [187]. Normal lifespan could be restored by either decreasing autophagy or by preventing PTP opening (CsA treatment, silencing of *ant-1.1* gene), indicating that mitochondrial permeabilization turns the normally protective process of autophagy into a death pathway [187], an event that could take place also in mammals [188]. A recent study in *Caenorhabditis elegans* demonstrated that OSCP downregulation in adult but not in developing worms was able to trigger PTP opening and the mitochondrial unfolded protein response causing decreased life span, which could be normalized by PTP inactivation by both silencing of the c-ring or peripheral stalk subunits (including subunit e) and treatment with CsA [189]. These results confirm the important role of the PTP in the aging process [108].

## 10. Species-Specific Differences as a Tool to Explore Mechanisms and Features of the PT

ANT proteins are highly conserved both in primary sequence and overall architecture, in the sense that the typical three homologous domains (which probably arose from duplication events) are conserved from yeast to mammals, implying that the duplication events occurred before the divergence of fungi, metazoa and plantae [190]. This sequence similarity—together with the lack of specific mutagenesis studies—does not allow an easy analysis of PTP species-specific features that can be referred to the ANTs. Extensive site-directed mutagenesis studies have instead explored the regulation and mechanisms of PTP formation from F-ATP synthases [40,76,82,94,137,191,192,193,194]. The F-ATP synthase catalytic domain is highly conserved from bacteria to mammals, with over 60% conservation of residues in subunit β [195]. In eukaryotes, the enzyme also contains “supernumerary” subunits that are anchored to the F_O_ region and include subunits e and g, which in yeast are strictly associated with dimers [196,197] and may mediate the association of monomers in mammals as well [198]. Structural data confirm that the central sector, including the c-ring, is widely conserved while extensive variations exist in the composition and structure of the peripheral stalk, particularly at the interface of dimers [84,199,200]. The PTP displays distinct properties in different species [99], including a characteristic mean conductance of 500 pS in mammals [40], 300 pS in *Saccharomyces cerevisiae* [135] and 53 pS in *Drosophila melanogaster* [148], where as already mentioned, the “PTP” appears to operate as a highly selective Ca^2+^ release channel [147]. What is the basis for these differences?

Mutations in subunit c affect the PTP [76,201]. Even if the hypothesis that the permeation pathway is provided by the c-ring turns out to be correct, the different conductance between mammals and yeast cannot be explained by the number of c subunits, hence by the size of the c-ring. Indeed, mammals have an 8-subunit c-ring that is smaller than the 10-subunit c-ring of *Saccharomyces cerevisiae* [66]. The c-ring may be necessary but not sufficient because prokaryotes do not undergo a PT in spite of the presence of a significantly larger c-ring than eukaryotes, with the record 17 c subunits of *Burkholderia pseudomallei* [66]. Furthermore, the PTP is drastically affected by three different modifications in the wedge region that do not affect c-ring assembly. (i) In *Saccharomyces cerevisiae*, the deletion of subunits e and g together with the α-helical portion of subunit b (which removes the wedge region) completely prevented the formation of a high-conductance channel [137]. (ii) Arginine 8 of subunit e and glutamate 83 of subunit g contribute to the stability of the e/g interaction by forming a salt bridge [193,202] and the substitution of arginine 8 in subunit e with alanine or glutamate or of glutamate 83 in subunit g with alanine or lysine drastically decreased PTP conductance, which could be rescued by simultaneous replacement of subunit e arginine 8 with glutamate and glutamate 83 of subunit g with lysine, most likely through reconstitution of the salt bridge [193]. (iii) Deletion of subunit g (which also caused depletion of subunit e) in HeLa cells completely prevented PTP channel opening in spite of the presence of an assembled c-ring [94].

Another intriguing set of observations points to the importance of the wedge region in imparting species-specific features to the PTP. Treatment of rat liver mitochondria with 2,3-butanedione or phenylglyoxal results in strong inhibition of the PTP through stable chemical modification of critical matrix arginine residue(s) [203]. The effect of phenylglyoxal on the PTP is species-specific, with an inhibitory effect in mitochondria from mice and yeast and an inducing effect in mitochondria from human cells and *Drosophila melanogaster* [192]. Given that in *Saccharomyces cerevisiae* arginine 107 of subunit g is the sole responsible for the effect of phenylglyoxal [192], by exploiting the phenotypic difference between human and yeast mitochondria, we have been able to show that expression of human subunit g in *Saccharomyces cerevisiae* confers the human phenotype to the yeast PT, with phenylglyoxal becoming now an inducer rather than an inhibitor [192].

An important contribution could come from subunit e (Figure 4). The GXXXG motif, which allows for tight packing with the α-helices of subunits g and b in the wedge region, is conserved both in terms of primary sequence and of position within the protein, and significant sequence similarity is also detectable in the transmembrane region. A specific region of interest is the C-terminal portion of the protein, which has been suggested to interact with the head group of the lipid in the outer face of the c-ring plug. Specifically, the involved lipid could be a lysolipid or a lipid with one very short acyl chain, which, at variance from the matrix side lipid, does not rotate with the c-ring [64]. The nature of the link between the C-terminus of subunit e and the lipid has not been defined.

The C-terminal lysine of subunit e is conserved in mammals and in zebrafish (*Danio rerio*), the PTPs of which are indistinguishable [204], while in *Drosophila melanogaster* the protein terminates with a histidine immediately following a lysine. *Caenorhabditis elegans, Saccharomyces cerevisiae* and *Yarrowia lipolytica* have unique C-termini ending with aspartic acid, threonine and alanine, respectively (Figure 4). Given that the yeast PTP appears to be similar to its mammalian counterpart and that a selective “PTP” is present in *Drosophila*, it is conceivable that the lysine residue may not be strictly required for the interaction of subunit e with the lipid plug and/or that binding can be contributed by multiple residues.

## 11. Summary and Conclusions

In this review, we have tried to assess the potential role(s) of the PT through the analysis of selected organisms in which our understanding of Ca^2+^ signaling, of the PTP and of the pathways to cell death has been studied in some detail. It soon became apparent to us that the available material varies widely between mammals, yeast, flies, crustaceans and worms, which makes for a difficult task. A good example is the case of *Caenorhabditis elegans*, an organism that allowed the elucidation of the pathways for apoptotic cell death in all kingdoms but where studies of the PT remain remarkably few, and that of crustaceans, where the lack of a PT remains without a mechanistic explanation, which is particularly striking given that mitochondria from these organisms take up Ca^2+^ and perform aerobic ATP synthesis with the same mechanism and through the same proteins as mammals and *Drosophila.* The demonstration that F-ATP synthase and ANT can form Ca^2+^-activated, high-conductance channels involved in the PT has been a mixed blessing, because it further complicated our efforts at comparing the potential role(s) of these proteins across species. Yet, by matching known features of the PTPs to mechanisms of cell death, in particular the presence of the mitochondrial pathway to apoptosis, we have outlined a working hypothesis that largely hinges on the presence of a PT in yeast, mammals and worms and its absence in crustaceans, and on what appears to be an intermediate case in *Drosophila melanogaster*, i.e., the presence of a Ca^2+^-selective Ca^2+^ release channel that shares some of the regulatory features of the *bona fide* PTP. Given that in mammals the PT provides a mechanism for the release of Ca^2+^, but can also trigger the release of cytochrome *c* and other proapoptotic proteins, we suggest that lack of a PT in crustaceans contributes to their resistance to anoxia (which is partly shared by *Drosophila*) and to dehydration, and that *Drosophila* has retained the Ca^2+^-release properties of the PTP but not its ability to permeabilize the inner membrane to solutes and to mediate the release of proapoptotic proteins. From an evolutionary perspective, it is interesting to note that there is no evidence for the activation of caspases by cytochrome *c* in non-vertebrates, suggesting that during evolution the apoptotic pathway appeared only once and that some features of the canonical pathways seen in higher species were lost in other species [205]. Consistent with this hypothesis, *Drosophila* DC3 and DC4 induce caspase activation in human cells, while human cytochrome *c* could not promote caspase activity in *Drosophila* cells [154]. It is therefore tempting to speculate that the PTP, which is also found in several plants [206], may have evolved differently in diverse species in the context of Ca^2+^ homeostasis and apoptosis in a process of molecular exaptation [206].

## Figures and Tables

**Figure 1 cells-12-01409-f001:**
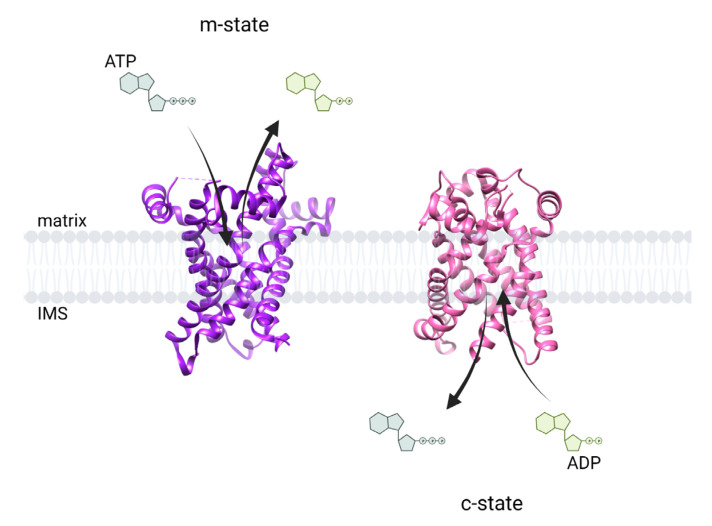
Structure of ANT in the m-state (*T. thermophila* PDB:6GCI) and c-state (*S. cerevisiae* PDB:4C9H). When ANT is in the m-state (purple) it is open from the matrix side and binds ATP (dark green) releasing ADP (light green). Contrarily, when ANT is in the c-state (pink) it is open from the intermembrane space (IMS) side and binds ADP releasing ATP. Arrows indicate the direction of adenine nucleotide transport.

**Figure 2 cells-12-01409-f002:**
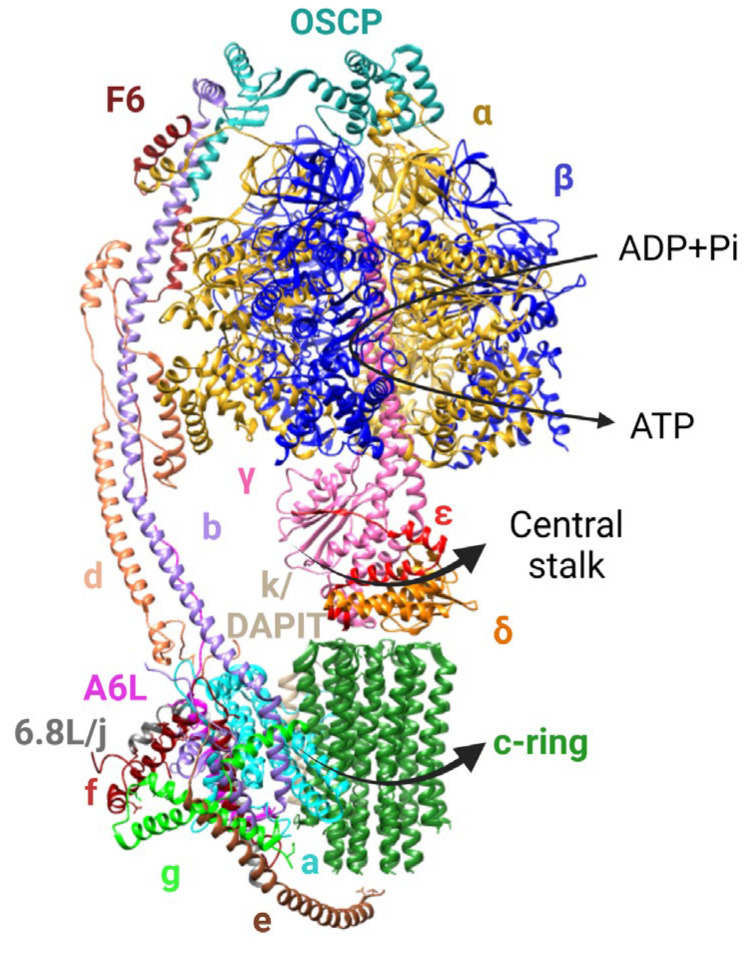
Structure of ovine monomeric F-ATP synthase (PDB:6TT7). Subunits are represented with a different color. Black thick arrows indicate the direction of rotation of the central stalk (subunits γ in pink, δ in orange and ε in red) and of the c-ring (dark green) during ATP synthesis. The upper thin arrow indicates where synthesis occurs.

**Figure 3 cells-12-01409-f003:**
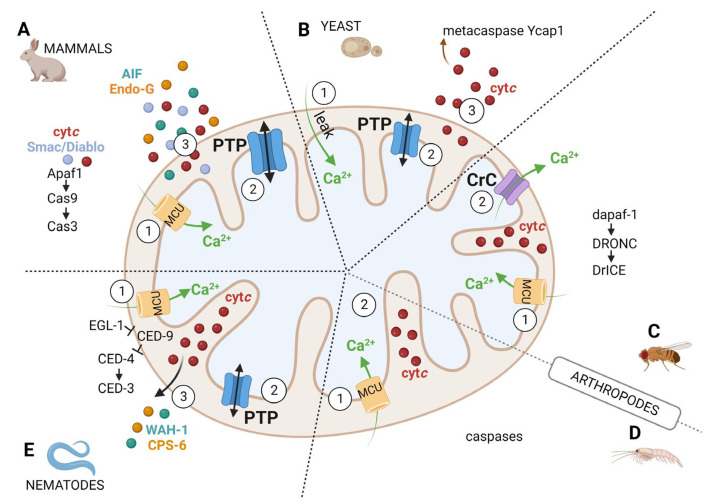
The PTP and caspase activation in mammals (**A**), *Saccharomyces cerevisiae* (**B**), *Drosophila melanogaster* (**C**), *Artemia franciscana* (**D**) and *Caenorhabditis elegans* (**E**). (1) Death stimuli induce mitochondrial Ca^2+^ uptake that can trigger PTP opening in mammals, yeast and worms and CrC opening in flies (2) but not in *Artemia*. PTP opening induces depolarization and (for long open times) swelling with rupture of the outer mitochondrial membrane and release of proapoptotic factors in mammals, yeast and worms (3). In *Drosophila*, where caspase activation appears not to require cytochrome *c*, opening of the CrC is not followed by swelling. PTP opening does not take place in *Artemia*. Endo-G, endonuclease G; cyt *c*, cytochrome *c*; MCU, mitochondrial Ca^2+^ uniporter; Cas, caspase; CrC, Ca^2+^-induced Ca^2+^ release channel; PTP, permeability transition pore.

**Figure 4 cells-12-01409-f004:**
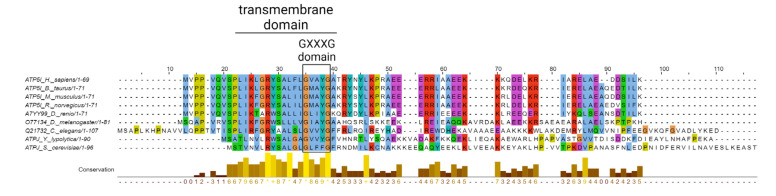
Alignment of subunits e from indicated species performed with Jalview software (according to Clustal sequence alignment program). The color scheme used for the alignment is default in Clustal X. The GXXXG domain and the predicted transmembrane domain (according to the TMHMM predictor) are indicated. The degree of conservation was computed with Jalview.

## Data Availability

No new data were created or analyzed in this study. Data sharing is not applicable to this article.

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
