# Peer review of "The Haves and Have-Nots: The Mitochondrial Permeability Transition Pore across Species"

_cells, 2023, doi:10.3390/cells12101409_

Round 1
Reviewer 1 Report
This is an excellent comprehensive review from the best experts in the field regarding the permeability transition pore (PTP) origin, structure, and function. The authors describe in detail the relationships between two major mechanisms, underlying the PTP, ANT- and ATP synthase-based, and discuss the current state of PTP research. The manuscript is very well written and easy to read. The cited literature is accurate and properly interpreted. The conclusions are supported by the cited literature and seem reasonable. Overall, this is a great review that undoubtedly will spark great interest among scientists studying the PTP.
Author Response
Thank you very much for the appreciation.
Reviewer 2 Report
The title of this review article suggests a comprehensive treatise about the function of the mitochondrial permeability transition pore mPTP, which in its biochemical architecture is, although intensively studied over many years, not finally solved and controversially discussed. Unfortunately, the expectation raised by the title and the abstract are not accomplished. Instead of a detailed description of findings from different studies, as they have been reviewed before, a meaningful general conclusion about the evolution of the PT in different taxa is expected from the title. Such a focus would be very interesting to a broad readership.
Here a few points for consideration by the authors:
1. The abstract does not provide any details about the outcome of the PT comparison of the selected species of this treatise. It only states that a comparison of the features of the PT “may be a useful exercise to help to understand the functions of the PT”. Here some clues should be provided.
2. While some biochemical aspects are provided in detail, the most interesting role of the mPTP in cell biological processes and their impact is not sufficiently covered. For instance, where does the PT play a role? Development, disease, aging? This is only marginally mentioned and not sufficiently addressed.
3. In their manuscript the authors compare the PT from a few species in some detail, occasionally mention some aspects from others, and ignore other species. What is the basis of this strategy?
4. Summary and conclusion: This part is rather limited in respect to the expectations raised in the title. Here some additional points may be included. For instance, what can be expected to come out from the comparison of the mPTP of different species? Conservation? Divergence?
5. Minor point: The captions of Figures 1, 2 and 4 is insufficient. The detailed information in these Figures needs to be provided here (in addition to the running text).
Author Response
Reviewer 2
The title of this review article suggests a comprehensive treatise about the function of the mitochondrial permeability transition pore mPTP, which in its biochemical architecture is, although intensively studied over many years, not finally solved and controversially discussed. Unfortunately, the expectation raised by the title and the abstract are not accomplished. Instead of a detailed description of findings from different studies, as they have been reviewed before, a meaningful general conclusion about the evolution of the PT in different taxa is expected from the title. Such a focus would be very interesting to a broad readership.
Answer: Thank you for the criticism. We acknowledge that there was a discrepancy between the title/abstract and the body of the review, which drifted somewhat from our original plan. This is due to the fact that the PT has been studied in detail only in a few species, which makes systematic comparisons difficult to make. An extreme example is posed by absence of the PT in Artemia and other crustaceans which in principle should not even have been covered; and by the features of the Drosophila channel, which sensu stricto is not a PTP. The idea was to elaborate from the latter examples to try and better understand the pathophysiology of the PTP. We have changed the title to “The haves and have-nots: The mitochondrial permeability transition across species”, which should now convey the appropriate message; and we have rewritten the abstract to better match the contents of the review. We hope that the scope of the review has now become clearer and that the overall frame is easier to understand.
Here a few points for consideration by the authors:
- The abstract does not provide any details about the outcome of the PT comparison of the selected species of this treatise. It only states that a comparison of the features of the PT “may be a useful exercise to help to understand the functions of the PT”. Here some clues should be provided.
Answer: The abstract has been rewritten.
- While some biochemical aspects are provided in detail, the most interesting role of the mPTP in cell biological processes and their impact is not sufficiently covered. For instance, where does the PT play a role? Development, disease, aging? This is only marginally mentioned and not sufficiently addressed.
Answer: The role of the PTP in disease and aging has been covered in so many reviews that we intentionally left it out, as was mentioned in the first paragraph of section 5.
- In their manuscript the authors compare the PT from a few species in some detail, occasionally mention some aspects from others, and ignore other species. What is the basis of this strategy?
Answer: We made a selection based on species where data on the PT could be compared with those on sensitivity to cell death (or resistance, as is the case of anoxia for Drosophila and Artemia). We tried to better clarify the logic in the revised manuscript.
- Summary and conclusion: This part is rather limited in respect to the expectations raised in the title. Here some additional points may be included. For instance, what can be expected to come out from the comparison of the mPTP of different species? Conservation? Divergence?
Answer: We hope that our short summary paragraph now fits the revised manuscript.
- Minor point: The captions of Figures 1, 2 and 4 is insufficient. The detailed information in these Figures needs to be provided here (in addition to the running text).
Captions were expanded as suggested.
Reviewer 3 Report
The manuscript by Frigo and colleagues is an excellent and timely review summarizing past and present knowledge of the mitochondrial permeability transition pore (PTP) in mammals, yeast, Drosophila, Artemia and C. elegans, and further exploring potential mechanisms of regulation of the PTP based on the differences observed in different species.
As already mentioned, it represents an excellent review, written by world leading experts in mitochondrial permeability transition and the role played by the ATP synthase in PTP formation.
Minor suggestions:
1.- In line 109, incorporate γ and δ subunits that are missing.
2.- In line 113, complete α3β3 complex, α and β are missing.
Author Response

(The authors gave the same response as above.)

Round 2
Reviewer 2 Report
The authors have addressed most of my suggestions to improve the manuscript (i.e., title, abstract, captions). However, I still see that this review would very much profit from an extension of the Summary and Conclusion section. As it stands now, it is rather a "Conclusion" section. The main general points from this review (summary) are not incorporated. I am missing general information about the evolutionary "conservation vs divergence" of the PT and its function in the selected species of this review. This is it what is important as a kind of 'take home message' to the reader. Moreover, addressing the question of why the PT has developed in the different species to fulfill different functions would be helpful. I strongly recommend to add this information prior to publication.
Author Response
We have expanded the last paragraph as suggested. We hope that this version adequately addressess your comments, thank you for asking (we like it better ourselves).